# Environment Diversification with Multi-head Neural Network for Invariant Learning

**Bo-Wei Huang    Keng-Te Liao    Chang-Sheng Kao    Shou-De Lin**
Department of Computer Science and Information Engineering
National Taiwan University, Taiwan
`{r10922007,d05922001,b07902046,sdlin}@csie.ntu.edu.tw`

## Abstract

Neural networks are often trained with empirical risk minimization; however, it has been shown that a shift between training and testing distributions can cause unpredictable performance degradation. On this issue, a research direction, invariant learning, has been proposed to extract causal features insensitive to the distributional changes. This work proposes EDNIL, an invariant learning framework containing a multi-head neural network to absorb data biases. We show that this framework does not require prior knowledge about environments or strong assumptions about the pre-trained model. We also reveal that the proposed algorithm has theoretical connections to recent studies discussing properties of variant and invariant features. Finally, we demonstrate that models trained with EDNIL are empirically more robust against distributional shifts.

## 1  Introduction

Ensuring model performance on unseen data is a common yet challenging task in machine learning. A widely adopted solution would be empirical risk minimization (ERM), where training and testing data are assumed to be independent and identically distributed. However, data in real-world applications can come with undesired biases, causing a shift between training and testing distributions. It has been known that the distributional shifts can severely harm ERM model performance and even cause the trained model to be worse than random predictions [10]. In this work, we focus on *Invariant Learning*, which aims at learning causal features expected to be robust against distributional changes. Invariant Risk Minimization (IRM) [1] has been proposed as a popular solution for invariant learning. Specifically, IRM is based on an assumption that training data are collected from multiple sources or *environments* having distinct data distributions. The learning objective is then designed as a standard ERM loss function with a penalty term constraining the trained model (e.g. classifier) to be optimal in all the environments.

IRM has shown to be effective; however, we note that IRM and many invariant learning methods rely on strict assumptions which limit the practical impacts. The limitations are summarized as follows.

**Prior Knowledge of Environments**    IRM assumes training data are collected from environments, and the environment labels (i.e. which data instance belongs to which environment) are given. However, the environment labels are often unavailable. Moreover, the definition of environments can be implicit, making human labeling more difficult and expensive. To find environments without supervision, Creager et al. [6] propose EIIL, a min-max optimization framework training the model of interest and inferring the environment labels. Another work, HRM [22] (or the extension, KerHRM [23]), parameterizes the environments and proposes clustering-based approaches to estimate the parameters. A recent method, ZIN [20], also learns to label data. However, it relies on carefully chosen features satisfying a series of theoretical constraints, and thus human efforts are still required.

36th Conference on Neural Information Processing Systems (NeurIPS 2022).

**Delicate Initialization**    EIIL is able to infer environments but requires an ERM model for initialization. Crucially, the ERM model should heavily depend on spurious correlations. Creager et al. [6] reveal that, for example, slightly underfitted ERM models may encode more spurious relationships in some cases. However, as the distributional shifts are assumed to be unknown in the training stage, appropriate initialization might be difficult to guarantee.

**Efficiency Issue**    HRM and KerHRM, though do not possess the above two limitations, suffer from the efficiency issue. Specifically, HRM is assumed to be trained with low dimensional data. As for KerHRM, although it extends HRM to avoid the issue of dimensions by adopting kernel methods, the computational costs of the proposed method can be very high if the data or model size is large.

This work proposes a novel framework, **E**nvironment **D**iversification with multi-head neural **N**etwork for **I**nvariant **L**earning (EDNIL). EDNIL is able to infer environment labels without supervision and achieve joint optimization of environment inference and invariant learning models. The underlying multi-head neural network explicitly diversifies the inferred environments, which is consistent with recent studies [5, 22, 23] revealing the benefits of diverse environments. Notably, the proposed neural network is functionally similar to a multi-class classifier and can be optimized efficiently. The advantages of EDNIL are summarized as:

- We implement this framework using various pre-trained models such as Resnet [14] and DistilBert [30], and evaluate it with diverse data types and varied biases. The results show that EDNIL can constantly outperform the existing state-of-the-art methods.

- The learning algorithm of EDNIL has theoretical connections to recent studies [5, 20, 22, 23] discussing conditions of ideal environments.

- EDNIL does not have the three limitations discussed above. The comparisons between EDNIL and other methods are shown in Table 1.

Table 1: A summary of the advantages of invariant learning methods.

|  | Unsupervised[1] | Insensitive Initialization | Efficiency |
|---|:---:|:---:|:---:|
| IRM [1] | ✗ | ✓ | ✓ |
| ZIN [20] | ✗ | ✓ | ✓ |
| EIIL [6] | ✓ | ✗ | ✓ |
| HRM [22] | ✓ | ✓ | ✗ |
| KerHRM [23] | ✓ | ✓ | ✗ |
| **EDNIL (Ours)** | ✓ | ✓ | ✓ |

## 2    Preliminaries and Related Works

The goal of EDNIL is to tackle out-of-distribution problems with invariant learning in the absence of manual environment labels. In Section 2.1, background knowledge about out-of-distribution generalization and invariant learning are introduced. In Section 2.2, we discuss recent studies investigating ideal environments. In Section 2.3, we introduce the existing unsupervised methods inferring environments.

### 2.1    Out-of-distribution Generalization and Invariant Learning

Following [1, 22], we consider a dataset $D = \{D^e\}_{e \in \text{supp}(\mathcal{E}_{\text{tr}})}$ with different sources $D^e = \{(x_i^e, y_i^e)\}_{i=1}^{n_e}$ collected under multiple training environments $e \in \text{supp}(\mathcal{E}_{\text{tr}})$. Random variable $\mathcal{E}_{\text{tr}}$ indicates the training environment labels. For simplicity, $X^e$, $Y^e$ and $P^e$ denote data, target label and distribution in environment $e$ respectively.

With $\mathcal{E}_{\text{all}}$ containing all possible environments such that $\text{supp}(\mathcal{E}_{\text{all}}) \supset \text{supp}(\mathcal{E}_{\text{tr}})$, the goal of out-of-distribution generalization is to learn a predictor $f(\cdot) : \mathcal{X} \to \mathcal{Y}$ as Equation 1, where $R^e(f) = \mathbb{E}_{X^e, Y^e}[l(f(X^e), Y^e)] = \mathbb{E}^e[l(f(X^e), Y^e)]$ measures the risk under environment $e$ with any loss

---

[1]A method is *unsupervised* if it does not require extra human efforts to obtain environments.

function $l(\cdot, \cdot)$. In general, for $e \in \mathrm{supp}(\mathcal{E}_{\mathrm{tr}})$ and $e' \in \mathrm{supp}(\mathcal{E}_{\mathrm{all}}) \setminus \mathrm{supp}(\mathcal{E}_{\mathrm{tr}})$, $P^{e'}(X, Y)$ is rather different from $P^e(X, Y)$.

$$f = \arg\min_f \max_{e \in \mathrm{supp}(\mathcal{E}_{\mathrm{all}})} R^e(f) \tag{1}$$

Recently, several studies [1, 4, 18, 26, 28] have attempted to tackle the generalization problems by discovering invariant relationships across all environments. A commonly proposed assumption is the existence of *invariant features* $X_{\mathrm{c}}$ and *variant features* $X_{\mathrm{v}}$. Specifically, raw features $X$ are assumed to be composed of $X_{\mathrm{c}}$ and $X_{\mathrm{v}}$, or $X = h(X_{\mathrm{c}}, X_{\mathrm{v}})$ where $h(\cdot)$ is a transformation function. Invariant features $X_{\mathrm{c}}$ are assumed to be equally informative for predicting targets $Y$ across environments $e \in \mathrm{supp}(\mathcal{E}_{\mathrm{all}})$. On the contrary, the distribution $P^e(Y|X_{\mathrm{v}})$ can arbitrarily vary across $e$. As a result, predictors depending on $X_{\mathrm{v}}$ can have unpredictable performance in unseen environments. In particular, the correlations between $X_{\mathrm{v}}$ and $Y$ are known as spurious and unreliable.

To extract $X_{\mathrm{c}}$, IRM [1] assumes there is an encoder $\Phi$ for obtaining representations $\Phi(X) \approx X_{\mathrm{c}}$. The encoder is trained with a regularization term enforcing simultaneous optimality of the predictor $w \circ \Phi$ in training environments, where dummy classifier $w = 1.0$ is a fixed multiplier for the encoder outputs:

$$\sum_{e \in \mathrm{supp}(\mathcal{E}_{\mathrm{tr}})} R^e(\Phi) + \lambda ||\nabla_{w|w=1.0} R^e(w \circ \Phi)||^2 \tag{2}$$

As $w$ is a dummy layer, the encoder $\Phi$ is also regarded as a predictor.

## 2.2 Ideal Environments

As $\mathcal{E}_{\mathrm{tr}}$ is unavailable or sub-optimal in most applications, learning to find appropriate environments (denoted by $\mathcal{E}_{\mathrm{learn}}$) is attractive. However, the challenge is the lack of knowledge of valid environments. Recently, Lin et al. [20] have proposed Equation 3 and 4 as the conditions of ideal environments, where $H$ is conditional entropy. To satisfy the conditions, Lin et al. [20] propose leveraging auxiliary information for model training. However, the method still requires extra human efforts to collect and verify the additional information.

$$H(Y|X_{\mathrm{c}}) = H(Y|X_{\mathrm{c}}, \mathcal{E}_{\mathrm{learn}}) \tag{3}$$

$$H(Y|X_{\mathrm{v}}) - H(Y|X_{\mathrm{v}}, \mathcal{E}_{\mathrm{learn}}) > 0 \tag{4}$$

Particularly, Equation 4 can be implied by empirical studies [5] where diversity of environments is recognized as the key to obtaining effective IRM models. To be more precise, large discrepancy of spurious correlations, or $P^e(Y|X_{\mathrm{v}})$, between environments is favored. As the environments give a clear indication of distributional shifts, IRM can easily identify and eliminate variant features. Beyond IRM, HRM [22] and KerHRM [23] can also be viewed as optimizing diversity via clustering-based methods specifically.

## 2.3 Unsupervised Environment Inference

Here we provide a detailed introduction of the existing environment inference methods that do not require extra human efforts. The general idea is to integrate environment inference with invariant learning algorithms who require provided environments.

EIIL [6] proposes formulating invariant learning as a min-max optimization problem. Specifically, EIIL is composed of two objectives, *Environment Inference* ($EI$) and *Invariant Learning* ($IL$), where $EI$ is optimized by maximizing the training penalty via labeling the data, and $IL$ is optimized by minimizing the training loss given the data labeled by $EI$. The two-stage framework bypasses the difficulty of defining environments; however, the training result heavily relies on the initialization of the $EI$ optimization. Specifically, the initialization demands a strongly biased ERM reference model; otherwise, EIIL can have a significantly weaker performance.

Another method, HRM [22], proposes a clustering-based method for learning plausible environments. HRM assumes that spurious correlations in each environment can be modeled by a parameterized function and the dataset is generated by the mixture of the functions. The parameters are then estimated by employing EM algorithm. Additionally, HRM equips a joint learning framework which alternatively learns invariant predictors and improves quality of clustering results. However, a known

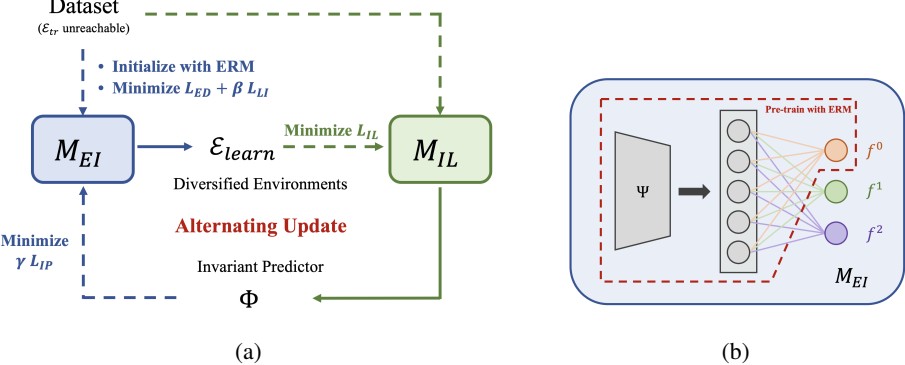

(a)                                    (b)

Figure 1: Concepts of EDNIL. (a) The joint learning framework of EDNIL. The environment inference model $M_{\text{EI}}$, containing a variant encoder $\Psi$ and environmental functions $f^e$, is trained with $L_{\text{EI}} = L_{\text{ED}} + \beta L_{\text{LI}} + \gamma L_{\text{IP}}$. The invariant learning model $M_{\text{IL}}$, containing an invariant predictor $\Phi$, is trained with $L_{\text{IL}}$. (b) The multi-head network structure of the environment inference model $M_{\text{EI}}$.

issue of HRM is an assumption that the data are represented by raw features. Data such as images and texts requiring non-linear neural networks to obtain representations are beyond the capability.

To extend HRM to a broader class of applications and improve the model performance, Liu et al. [23] propose KerHRM. The main idea is to adopt the Neural Tangent Kernel [15] method which transforms non-linear neural network training into a linear regression problem on the proposed Neural Tangent Features space. As a result, KerHRM elegantly resolves the shortcomings of HRM and is shown to be more effective. However, the proposed method and its implementations bring additional computational costs depending on data and model capacity. For applications favoring large datasets and pre-trained models, such as Resnet [14] and BERT [7], KerHRM may not be an affordable option at the present stage.

## 3 Methodology

In this section, we propose a general framework to learn invariant model without manual environment labels. As shown in Figure 1a, our proposed method consists of two models, $M_{\text{EI}}$ and $M_{\text{IL}}$. Given the pooled data $(X, Y)$, $M_{\text{EI}}$ infers environments $\mathcal{E}_{\text{learn}}$ satisfying Condition 3 and 4, and $M_{\text{IL}}$ is an invariant model trained with the inferred environments. Our framework is jointly optimized with alternating updates. The learned $M_{\text{IL}}$ can provide information of invariant features to $M_{\text{EI}}$, so that Condition 3 and 4 can be fulfilled simultaneously. Note that $M_{\text{EI}}$ only serves at train time to provide environments for invariant learning. At test time, only $M_{\text{IL}}$ is needed to perform invariant predictions.

### 3.1 The Environment Inference Model

The target of environment inference is to partition data with environment labels $\mathcal{E}_{\text{learn}}$ satisfying Condition 3 and 4. In this regard, we propose a graphical model, which is a sufficient condition for Condition 3 and 4 (the proof is in Appendix A), as our foundation of inference model and learning objectives. The graph is shown in Figure 2, where the data generation process follows the proposed example in [1].

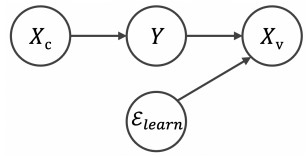

Figure 2: The graphical model.

The inference model, $M_{\text{EI}}$, is a neural network learning to realize the underlying graphical model. Following the idea of parameterizing environments from HRM [22] and KerHRM [23], we assume the distinct mapping relation between $X$ and $Y$ in environment $e$ can be modeled by a function $f^e(\Psi(X))$, where $\Psi(X)$ is learned representations expected to encode variant features $X_{\text{v}}$ and $f^e$ is an environmental function responsible for predicting $Y$. Instead of employing clusters, we propose building a multi-head neural network as shown in Figure 1b; particularly, a single-head network in $M_{\text{EI}}$ with shared parameters corresponds to a cluster center in HRM or KerHRM. The training procedure of $M_{\text{EI}}$ can be divided into *inference stage* and *learning stage*.

### 3.1.1 Inference Stage of $M_{\text{EI}}$

The goal is to infer an environment label for each training data. As in the graphical model, $\mathcal{E}_{\text{learn}}$ is associated with variant relationships. Inspired by multi-class classification problem, we propose Equation 5, where the probability $P(e \mid X, Y)$ is estimated via a softmax of negative $l(f^e(\Psi(X)), Y)$ divided by a constant temperature $\tau$. The function $l$ is expected to be the commonly used cross entropy or mean squared error that measures the discrepancy between $Y$ and $f^e(\Psi(X))$ for each environment $e$. Intuitively, each data prefers the environment whose model has better prediction.

$$P(\mathcal{E}_{\text{learn}} = e \mid X, Y) = \frac{\exp\left(-l(f^e(\Psi(X)), Y)/\tau\right)}{\sum\limits_{e' \in \text{supp}(\mathcal{E}_{\text{learn}})} \exp\left(-l(f^{e'}(\Psi(X)), Y)/\tau\right)} \tag{5}$$

### 3.1.2 Learning Stage of $M_{\text{EI}}$

The goal is to update the neural network to improve the quality of inference. Based on the structure of the graphical model, three losses are designed for minimization, i.e. *Environment Diversification Loss* ($L_{\text{ED}}$), *Label Independency Loss* ($L_{\text{LI}}$) and *Invariance Preserving Loss* ($L_{\text{IP}}$). In particular, $L_{\text{ED}}$ and $L_{\text{IP}}$ correspond to the concepts of Condition 4 and 3 respectively.

**Environment Diversification Loss ($L_{\text{ED}}$)**    We consider maximizing $H(Y|X_{\text{v}}) - H(Y|X_{\text{v}}, \mathcal{E}_{\text{learn}})$ to satisfy Condition 4 and capture more diverse variant relationships. Given the estimated $P(\mathcal{E}_{\text{learn}}|X, Y)$, $L_{\text{ED}}$ selects the most probable environment and its corresponding network for optimization:

$$L_{\text{ED}} = -\sum_i w_i \max_e [\log P(e \mid x_i, y_i)] \tag{6}$$

For each data $(x_i, y_i)$, although only one environment $e_i = \arg\max_e P(e \mid x_i, y_i)$ is selected for the minimization, the softmax simultaneously propagates gradient to maximize $l(f^{e'}(\Psi(x_i)), y_i)$ for $e' \neq e_i$. The network learns to maximize the dependency between $\mathcal{E}_{\text{learn}}$ and $Y$ given variant representations. In terms of spurious correlations, the distinction between environments is expected to become clearer accordingly. In practice, we utilize scaling weight $w_i$ inversely proportional to the size of $e_i$. The importance of smaller environments will be thus enhanced within the summation.

**Label Independency Loss ($L_{\text{LI}}$)**    With d-separation [25], $\mathcal{E}_{\text{learn}}$ is independent of $Y$ in the graphical model. Hence, $L_{\text{LI}}$ constraints their dependency measured by the mutual information $I(Y; \mathcal{E}_{\text{learn}})$. Empirically, $L_{\text{LI}}$ prevents a trivial solution that environments are determined purely by target labels $Y$ regardless of input features $X$, which is undesirable for invariant learning. To minimize $I(Y; \mathcal{E}_{\text{learn}})$, it can be verified that it is equivalent to minimizing Equation 7 given $P(Y)$ is known.

$$L_{\text{LI}} = \mathbb{E}_{e \sim P(\mathcal{E}_{\text{learn}})}[\sum_y P(y|e) \log P(y|e)] \tag{7}$$

**Invariance Preserving Loss ($L_{\text{IP}}$)**    For Condition 3, as $M_{\text{IL}}$ learns some invariant relationships after several training steps (Section 3.2), $L_{\text{IP}}$ can be considered to exclude invariant features from the diversification. Specifically, designed as the contrary of $L_{\text{ED}}$, $L_{\text{IP}}$ limits the variance of expected loss from invariant predictor $\Phi$ (in $M_{\text{IL}}$) across environments (Equation 8). Similar idea can be found in [17]. However, instead of training invariant model given known environments, we freeze the invariant predictor and regularize the adjustment of $\mathcal{E}_{\text{learn}}$ (i.e. the updates of $M_{\text{EI}}$) here.

$$L_{\text{IP}} = \text{Var}_{e \sim P(\mathcal{E}_{\text{learn}})}[\mathbb{E}^e[l(\Phi(X^e), Y^e)]] \tag{8}$$

In summary, the training loss of $M_{\text{EI}}$ can be summarized as *Environment Inference Loss* ($L_{\text{EI}}$). The regularization strengths of $L_{\text{LI}}$ and $L_{\text{IP}}$ can be controlled by hyper-parameters $\beta$ and $\gamma$ respectively:

$$L_{\text{EI}} = L_{\text{ED}} + \beta L_{\text{LI}} + \gamma L_{\text{IP}} \tag{9}$$

In addition, before minimizing $L_{\text{EI}}$, we pre-train our $\Psi$ and one arbitrary $f^e$ with ERM. In general, it empirically facilitates better feature extraction. Unlike EIIL [6] taking ERM as a reference model heavily relying on variant features, EDNIL performs more consistently under various choices of ERM. Namely, the initialization of EDNIL can be more arbitrary than that of EIIL. We verify the argument in Section 4.

## 3.2 The Invariant Learning Model

To identify invariance across environments, IRM [1] is selected as our base algorithm optimizing the invariant predictor $\Phi$ in our model $M_{\text{IL}}$. As for the required environment partitions during training, we assign environment label $e \in \text{supp}(\mathcal{E}_{\text{learn}})$ with largest $P(e|x_i, y_i)$, inferred by $M_{\text{EI}}$ (Section 3.1.1), to each data $(x_i, y_i)$. However, it is inevitable that there exist some noises in automatically inferred environments, especially in the beginning of joint optimization. To reduce the impact of immature environments on invariant learning, we calculate the confidence score $c_e$ for each environment $e \in \text{supp}(\mathcal{E}_{\text{learn}})$, i.e. $\mathbb{E}^e[P(e|X^e, Y^e)]$. Our training objective is modified to minimize *Invariant Learning Loss* ($L_{\text{IL}}$) that considers the weighted average of environmental losses:

$$L_{\text{IL}} = \sum_{e \in \text{supp}(\mathcal{E}_{\text{learn}})} w_e \cdot [R^e(\Phi) + \lambda ||\nabla_{w|w=1.0} R^e(w \circ \Phi)||^2] \qquad (10)$$

$$w_e = \frac{c_e}{\sum_{e' \in \text{supp}(\mathcal{E}_{\text{learn}})} c_{e'}} \qquad (11)$$

# 4 Experiments

We empirically validate the proposed method on biased datasets, Adult-Confounded, CMNIST, Waterbirds and SNLI. The generation of spurious correlations mainly follow the protocols proposed by [1, 6, 9, 29]. In Section 4.1, Adult-Confounded and CMNIST are tested with Multilayer Perceptron (MLP). In Section 4.2, two more complex datasets, Waterbirds and SNLI, are taken for evaluating the integration of transfer learning. Deep pre-trained models will be fine-tuned to discover variant and invariant representations.

The following four methods are selected as our competitors: Empirical Risk Minimization (ERM), Environment Inference for Invariant Learning (EIIL [6]), Kernelized Heterogeneous Risk Minimization (KerHRM [23]) and Invariant Risk Minimization (IRM [1], Equation 2). EIIL and KerHRM are invariant learning methods with unsupervised environment inference, which share the same settings as EDNIL. HRM [22] is replaced by KerHRM for non-linearity. For IRM who requires environment partitions, we re-label $\mathcal{E}_{\text{oracle}}$ on each given biased training set, which diversifies the spurious relationships to elicit upper-bound performance of IRM.

For hyper-parameter tuning, we split 10% of training data to construct an in-distribution validation set. In each dataset, several testing environments with different distributions are listed to evaluate the robustness of each method, and we mainly take worst-case performance for assessment. As all tasks in this section are classification problems, accuracy is adopted as the evaluation metric.

Besides, more experimental details are revealed in Appendix B. We also discuss additional experiments in Appendix C, including solutions to regression problem and model stability given different spurious strengths at train time.

## 4.1 Simple Datasets with MLP

This section includes two simple datasets, Adult-Confounded and CMNIST, where spurious correlations are synthetically produced with the predefined strengths. For all competitors, MLP is taken as the base model and full-batch training is implemented. Since KerHRM performs unstably over random seeds, we first average the results after 10 runs as its first score, and select top 5 among them as the second one, which will be marked with an asterisk (*) in each table.

### 4.1.1 Discussions on Adult-Confounded

We take UCI Adult [16] to predict binarized income levels (above or below $50,000 per year) [2]. Following [6], individuals are re-sampled according to sensitive features *race* and *sex* to simulate spurious correlations. Specifically, with binarized *race* (Black/Non-Black) and *sex* (Female/Male), four possible subgroups are constructed: Non-black Males (SG1), Non-black Females (SG2), Black Males (SG3), and Black Females (SG4). Keeping original train/test split and subgroup sizes from

---

[2] UCI Adult dataset is widely used in algorithmic fairness papers. However, a recent study [8] discusses some limitations of this dataset, such as the choice of income threshold.

Table 2: $P(Y = 1|SG)$ for Adult-Confounded. IID shares spurious correlations with the train set. IND has no bias on *race* and *sex*. OOD defines the worst-case performance.

|  | Train | Test (IID) | Test (IND) | Test (OOD) |
|---|---|---|---|---|
| SG1 | 0.9 | 0.9 | 0.5 | 0.1 |
| SG2 | 0.1 | 0.1 | 0.5 | 0.9 |
| SG3 | 0.9 | 0.9 | 0.5 | 0.1 |
| SG4 | 0.1 | 0.1 | 0.5 | 0.9 |

Table 3: Testing accuracy (%) on Adult Confounded. Three subsets are defined in Table 2. $\beta = 0$ and $\gamma = 0$ indicate the removal of $L_{\mathrm{LI}}$ and $L_{\mathrm{IP}}$ when training $M_{\mathrm{EI}}$.

|  | IID | IND | **OOD** |
|---|---|---|---|
| ERM | **92.4** $\pm$ 0.1 | 66.8 $\pm$ 0.3 | 40.7 $\pm$ 0.5 |
| EIIL | 76.2 $\pm$ 0.4 | 73.5 $\pm$ 0.5 | 70.2 $\pm$ 1.7 |
| KerHRM | 82.4 $\pm$ 3.9 | 75.1 $\pm$ 4.0 | 67.9 $\pm$ 9.3 |
| KerHRM$^*$ | 81.2 $\pm$ 1.8 | 78.5 $\pm$ 0.3 | 75.6 $\pm$ 1.9 |
| EDNIL | 80.7 $\pm$ 0.4 | **79.1** $\pm$ **0.4** | **77.5** $\pm$ **0.3** |
| EDNIL$_{\beta=0}$ | 91.8 $\pm$ 0.0 | 66.7 $\pm$ 0.1 | 41.3 $\pm$ 0.7 |
| EDNIL$_{\gamma=0}$ | 78.2 $\pm$ 2.4 | 75.4 $\pm$ 1.6 | 72.5 $\pm$ 3.3 |
| IRM | 79.9 $\pm$ 0.4 | 79.3 $\pm$ 0.3 | 78.8 $\pm$ 0.4 |

UCI Adult, we sample data based on the given target distributions in each sensitive subgroup as shown in Table 2. In particular, OOD contributes the worst-case performance to validate if the predictions rely on group information. In this task, MLP with one hidden layer of 96 neurons is considered. For IRM, four environments comprise $\mathcal{E}_{\mathrm{oracle}}$, where the correlations between variant features ($race$, $sex$) and target $Y$ are distributed without overlapping. More details are provided in Appendix B.

**Results**  The results are shown in Table 3. With strong spurious correlations at train time, ERM obtains high accuracy as the correlations remain aligned; however, its generalization to other testing distributions is limited. Among all invariant learning methods without prior environment labels, EDNIL can perfectly identify variant features and infer ideally diversified environments. Therefore, it achieves the most invariant performance over different testing distributions. EIIL can improve consistency to some degree, but not as strong as EDNIL. A possible reason is that empirically trained reference model is not guaranteed to be purely variant [6]. For KerHRM, it performs inconsistently across random seeds, which is reflected in the large standard deviation. In some cases, the performance hardly improves over iterations, as observed by Liu et al. [23].

**Ablation Study for $L_{\mathrm{EI}}$**  We first claim the importance of $L_{\mathrm{LI}}$, which constraints label dependency, by setting the coefficient $\beta$ to zero. As discussed in Section 3, the resulting environments are determined purely by target labels, and thus lead to inferior performance for invariant learning as shown in Table 3. Next, we demonstrate the effectiveness of joint optimization in Figure 3a. The regularization $L_{\mathrm{IP}}$ promotes environment inference, so that the worst-case performance improves and remains stable over iterations. According to Table 3, if the coefficient $\gamma$ is turned off, feedback generated by $M_{\mathrm{IL}}$ will be ignored and the effect of invariant learning risks being undesirable.

### 4.1.2 Discussions on CMNIST

We report our evaluation on a noisy digit recognition dataset, CMNIST. Following [1], we first assign $Y = 1$ to those whose digits are smaller than 5 and $Y = 0$ to the others. Next, we apply label noise by randomly flipping $Y$ with probability 0.2. Finally, the digits are colored with color label $C$, which is generated by randomly flipping $Y$ with probability $e$. For training, two equal-sized environments with $e = 0.1$ and $e = 0.2$ are merged, which is equivalent to one with $e = 0.15$ on average. For testing, three situations are considered when $e$ is 0.1, 0.5 or 0.9, respectively. Note that when $e = 0.1$, the spurious correlation is much aligned with the training set. On the other hand, $e = 0.9$ defines the most challenging case since the spurious correlation shifts most dramatically from training.

For all competitors except KerHRM, we select MLP with two hidden layers of 390 neurons, and consider the whole dataset (50,000 samples) for training. For KerHRM who requires massive computing resources, we follow the settings recommended by [23]. Specifically, we randomly select 5,000 samples and train MLP with one hidden layer of 1024 neurons. To construct ideally diversified $\mathcal{E}_{\mathrm{oracle}}$ for IRM, we pack all examples with $C = Y$ into one environments, and $C \neq Y$ into the other.

**Results**  The results are shown in Table 4. First of all, not surprisingly ERM still adopts poorly to distributional shifts. Among all invariant learning methods without manual environment labels,

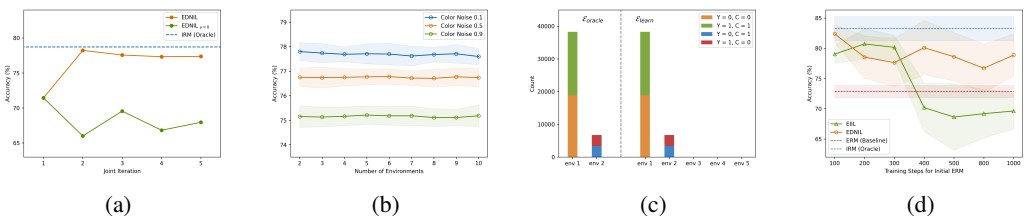

(a)                  (b)                  (c)                  (d)

Figure 3: Analysis results. (a) Ablation study of joint optimization on Adult-Confounded, where $\gamma = 0$ means the removal of $L_{\text{IP}}$. (b) CMNIST Performance of EDNIL with different configured numbers of environments. (c) Comparison between $\mathcal{E}_{\text{learn}}$ inferred by EDNIL and the oracle environments $\mathcal{E}_{\text{oracle}}$ on CMNIST. (d) Testing EIIL's and EDNIL's sensitivity to initialization on Waterbirds.

EDNIL gets closest to IRM with $\mathcal{E}_{\text{oracle}}$, achieving consistent and robust performance in this dataset. As shown in Figure 3c, EDNIL provides almost ideally diversified $\mathcal{E}_{\text{learn}}$ for invariant learning.

**Number of Environments** As shown in Figure 3b, EDNIL is not sensitive to the predefined number of environments. Specifically, when the environment number is larger than the oracle (i.e. 2), some environment classifiers become redundant. Each of them provides a moderate constant loss, taking up fixed and ignorable space in the softmax function. The visualization of $\mathcal{E}_{\text{learn}}$ with 5 available environments is shown in Figure 3c. Additionally, training $M_{\text{EI}}$ in EDNIL with more environments is much more efficient than clustering-based methods, such as the one proposed in KerHRM.

Table 4: Testing accuracy (%) of CMNIST, where color noise 0.9 makes the worst-case environment.

| Color Noise | 0.1 | 0.5 | **0.9** |
|---|---|---|---|
| ERM | **88.4** $\pm$ **0.3** | 55.0 $\pm$ 0.5 | 21.7 $\pm$ 0.8 |
| EIIL | 79.6 $\pm$ 0.3 | 71.7 $\pm$ 0.7 | 63.1 $\pm$ 0.5 |
| KerHRM | 74.3 $\pm$ 0.7 | 66.2 $\pm$ 1.7 | 58.0 $\pm$ 11.5 |
| KerHRM* | 71.3 $\pm$ 0.7 | 68.5 $\pm$ 0.5 | 66.1 $\pm$ 0.7 |
| EDNIL | 77.7 $\pm$ 0.4 | **76.8** $\pm$ **0.3** | **75.2** $\pm$ **0.4** |
| IRM | 77.8 $\pm$ 0.4 | 76.8 $\pm$ 0.4 | 75.2 $\pm$ 0.3 |

## 4.2 Complex Datasets with Pre-trained Deep Learning Models

This section extends MLP to deep learning models with pre-trained weights for more complex data. With mini-batch fine-tuning, we consider all competitors but exclude KerHRM due to the efficiency issue. In Section 4.2.1, image dataset, Waterbirds [29], with controlled spurious correlations is selected for evaluating the generalization on more high-dimensional images. In Section 4.2.2, a real-world NLP dataset, SNLI [3], is considered. The biases in SNLI are naturally derived from the procedure of data collection, and we define biased subsets for evaluation following Dranker et al. [9].

### 4.2.1 Discussions on Waterbirds

In Waterbirds [29], each bird photograph, from CUB dataset [31], is combined with one background image, from Places dataset [33]. Both birds and backgrounds are either from land or water, and our target is to predict the binarized species of birds. At train time, landbirds and waterbirds frequently present in land and water backgrounds respectively. Therefore, empirically trained models are prone to learn context features, and fail to generalize as background varies [2, 6, 10, 21, 29].

To split a validation set whose overall distribution is i.i.d. to the training set, we merge original training and validation data [3] and split 10% for hyper-parameter tuning. For testing, we observe all four combinations of birds and backgrounds in the original testing set. Among them, the minor subgroup (waterbirds on land) contributes the most challenging case. In this task, Resnet-34 [14] is chosen for mini-batch fine-tuning. Given two binary labels $(target, background)$, we distribute $target = background$ and $target \neq background$ into two different environments and apply balanced class weights for the oracle settings of IRM.

**Results** The results are shown in Table 5. As observed in [6, 29], ERM suffers in the hardest case (i.e. waterbirds on land). EIIL also performs poorly in this case. With a more sophisticated learning

---
[3]In the original training split, backgrounds are unequally distributed in each class. However, in the original validation split, they are equally distributed, which is not i.i.d. to the training.

Table 5: Testing accuracy (%) of Waterbirds, where Y and BG means target and background respectively. The subgroup (Water, Land) contributes the worst-case performance.

| (Y, BG) | (Land, Land) | (Water, Water) | (Land, Water) | **(Water, Land)** |
|---------|--------------|----------------|---------------|-------------------|
| ERM | 99.4 $\pm$ 0.0 | **91.4** $\pm$ **0.2** | **90.9** $\pm$ **0.8** | 72.8 $\pm$ 1.0 |
| EIIL | **99.4** $\pm$ **0.3** | 90.5 $\pm$ 1.8 | 89.3 $\pm$ 3.7 | 68.6 $\pm$ 5.4 |
| EDNIL | 98.5 $\pm$ 0.6 | 89.9 $\pm$ 1.5 | 90.3 $\pm$ 3.0 | **78.6** $\pm$ **4.3** |
| IRM | 98.0 $\pm$ 0.5 | 90.6 $\pm$ 1.1 | 89.5 $\pm$ 1.7 | 83.2 $\pm$ 2.2 |

framework, EDNIL narrows the gaps between subgroups and raises the worst-case performance. The results show that EDNIL is more resistant to distributional shifts.

**Choice of Initialization**    Both EIIL and EDNIL take ERM as initialization. As mentioned in Section 1, heavy dependency on initialization is risky when testing distribution is unavailable. Therefore, we take ERM with different training steps for EIIL and EDNIL to verify the stability. The results are shown in Figure 3d. As suggested by [6], EIIL works only with underfitted reference model in this case. If the reference model is more well-trained, the performance of EIIL will greatly decline since ERM might get distracted from variant features. One is prone to be misled into an undesirable choice for EIIL when seeking hyper-parameters without prior knowledge of distributional shifts. For instance, the validation score of EIIL with 500-step reference model (95.8%) is higher than that with 100-step (94.7%), which is not consistent with their performances on testing. In comparison, EDNIL performs more consistent across different pre-training steps, which accentuates our strength of less sensitive initialization.

### 4.2.2  Discussions on SNLI

The target of SNLI [3] is to predict the relation between two given sentences, premise and hypothesis. Recent studies [12, 24, 27] reveal hypothesis bias in SNLI, which is characterized by patterns in hypothesis sentences highly correlated with a specific label. One can achieve low empirical risk without considering premises during prediction. However, as the bias no longer holds, the performance degradation occurs [11, 24].

We sample 100,000 examples and consider all classes, *entailment*, *neutral* and *contradiction*, for our experiment. Following [9], we define three subsets, unbiased, bias aligned and bias misaligned, by training a biased model with hypothesis as its only input. The specification of the subsets is as follows:

- Unbiased: Examples whose predictions from the biased model are ambiguous
- Aligned: Examples that the biased model can predict correctly with high confidence
- Misaligned: Examples that the biased model can predict incorrectly with high confidence

The proportions of the three subsets are 17%, 67% and 16%, respectively. Due to the minority, the bias misaligned subset is more likely to be ignored and thus defines the worst-case performance.

For all methods, DistilBERT [30] is taken as the pre-trained model for further mini-batch fine-tuning. For $\mathcal{E}_{\text{oracle}}$, we assign the bias aligned subset to the first environment, and the bias misaligned subset to the second. In order to make bias prevalence equal in the two environments, unbiased samples are scattered proportionally to the two environments.

Table 6: Testing accuracy (%) on SNLI, where the misaligned defines the worst-case performance.

| Subset | Unbiased | Aligned | **Misaligned** |
|--------|----------|---------|----------------|
| ERM | **74.6** $\pm$ **0.3** | 94.7 $\pm$ 0.2 | 52.6 $\pm$ 0.9 |
| EIIL | 74.2 $\pm$ 0.3 | **95.0** $\pm$ **0.1** | 51.7 $\pm$ 1.3 |
| EDNIL | 74.3 $\pm$ 0.8 | 94.2 $\pm$ 0.2 | **54.5** $\pm$ **1.0** |
| IRM | 74.0 $\pm$ 0.9 | 92.3 $\pm$ 0.5 | 56.9 $\pm$ 1.1 |

**Results**    The results are shown in Table 6. As reported by [9], ERM receives higher score on the bias aligned subset, but it fails in the bias misaligned case. Among all invariant learning methods without environment labels, only EDNIL improves on the bias misaligned subset. Namely, even though the definitions of biases are at a high level, EDNIL is still capable of encoding and diversifying possible variant features.

# 5 Limitations

Our learning algorithm for environment inference is based on the graphical model plotted in Figure 2. As data are not necessarily generated by the presumed process, there can exist biases that cannot be captured by the proposed neural network. In the paper, we provide empirical studies of effectiveness on diverse datasets, while we are still aware that a stronger guarantee of performance is required.

# 6 Conclusions and Societal Impacts

This work proposes EDNIL for training models invariant to distributional shifts. To infer environments without supervision, we propose a multi-head neural network structure to identify and diversify plausible environments. With joint optimization, the resulting invariant models are shown to be more robust than existing solutions on data having distinct characteristics and different strengths of biases. We attribute the effectiveness to the underlying learning objectives, which are consistent with recent studies of ideal environments. Additionally, we note that the classifier-like structure of environment inference model makes EDNIL easy to combine with off-the-shelf pre-trained models and trained more efficiently.

Our contributions to invariant learning have broader societal impacts on numerous domains. For instance, it can encourage further research and real-world applications on debiasing machine learning systems. To be more specific, the identification and elimination of potential biases can facilitate more robust model training. It can be beneficial to many real applications where distributional shifts commonly occur, such as autonomous driving, social media and healthcare.

Furthermore, as discussed by [6], invariant learning can promote algorithmic fairness in some ways. In particular, our empirical achievements on Adult-Confounded can prevent discrimination against sensitive demographic subgroups in decision-making process. It shows that EDNIL has a potential to learn a fair predictor without prior knowledge of sensitive attributes, which is related to [13, 19, 32]. We expect that one can extend our work to more fairness benchmarks and criteria in the future.

Last but not least, it is worth mentioning some cautions, however. Since the invariant learning algorithm claims to find invariant relationships, one might cast more attention on feature importance of the invariant model and even incorporate the results into further research or applications. Nevertheless, the results are reliable only when the model is trained appropriately. Insufficient data collection or careless training process, for example, can certainly affect the identification of invariant features, and thus mislead experimental findings. As a result, we believe that adequate and careful preparations and analyses are essential before drawing conclusions from the inferred invariant relationships.

## Acknowledgments and Disclosure of Funding

We would like to thank the anonymous reviewers for their helpful suggestions. This material is based upon work supported by National Science and Technology Council, ROC under grant number 111-2221-E-002 -146 -MY3 and 110-2634-F-002-050 -.

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
