# Appendix

## A   Proof of the Underlying Graphical Model of EDNIL

We assume data are generated by Equation 12 which is the process adopted by Arjovsky et al. [1] and Lin et al. [20], where $g_c$, $g_v$ are deterministic functions, and $\epsilon_c$, $\epsilon_v$ are random noises independent of $X_c$ and $X_v$.

$$Y = g_c(X_c, \epsilon_c),$$
$$X_v = g_v(Y, \epsilon_v). \tag{12}$$

Equation 12 establishes a chain that $X_c \rightarrow Y \rightarrow X_v$. With the additional relation $\mathcal{E} \rightarrow X_v$ proposed in our graphical model, the following two dependencies can be obtained via d-separation [25]:

$$Y \perp\!\!\!\perp \mathcal{E} \mid X_c,$$
$$Y \not\perp\!\!\!\perp \mathcal{E} \mid X_v. \tag{13}$$

As can be seen, the conditional independence between $Y$ and $\mathcal{E}$ given $X_c$ leads to $H(Y \mid X_c) = H(Y \mid X_c, \mathcal{E})$. On the other hand, the dependency between $Y$ and $\mathcal{E}$ given $X_v$ implies that there exists an environment variable satisfying $H(Y \mid X_v) > H(Y \mid X_v, \mathcal{E})$.

## B   Experimental Details

### B.1   Implementation Resources

Our implementations of EDNIL are in the repository [4].

All experiments were run on a GeForce RTX 3090 machine. The training time and GPU memory consumption of EDNIL are specified in Table 7. It takes approximately 30 hours for EDNIL to accomplish all tasks, including main experiments and analyses.

For the choices of deep encoders, we utilize Resnet34 from torchvision [5] on Waterbirds, and DistilBERT from Huggingface [6] on SNLI. Both network architectures and pre-trained weights are kept as the default.

Table 7: Training time and GPU memory consumption of EDNIL (per run).

|            | Adult-Confounded | CMNIST  | Waterbirds | SNLI     |
| ---------- | ---------------- | ------- | ---------- | -------- |
| Time       | 1 min            | 2 min   | 40 min     | 60 min   |
| GPU memory | 1.3 GiB          | 1.6 GiB | 7.0 GiB    | 10.2 GiB |

### B.2   Hyper-parameter Tuning

Without leaking into out-of-distribution information, 10% of training data are split as an in-distribution validation set. Particularly, we infer environments on the validation data with $M_{\text{EI}}$ and determine hyper-parameters according to the worst-environment score.

For $M_{\text{EI}}$, we select number of environments from 2 to 5, $\tau$ in softmax function from 0.05 to 0.5, $\beta$ and $\gamma$ in $L_{\text{EI}}$ from 0.2 to 10. In $L_{\text{ED}}$, we clip overaggressive $w_i$ with an upper bound $w_{\text{thres}}$ for training stability, and it is chosen from 1.2 to 5. As for $M_{\text{IL}}$, the penalty strength $\lambda$ in $L_{\text{IL}}$ is selected from {2, 10, 100, 1000}. Following Arjovsky et al. [1], we conduct an annealing mechanism before using the configured penalty strength. The chosen number of annealing iterations ranges from 20% to 80% of the whole. For the complex datasets, we consider a longer annealing period (larger than 50%) to learn basic representations better.

---

[4]https://github.com/joe0123/EDNIL
[5]https://pytorch.org/hub/pytorch_vision_resnet
[6]https://huggingface.co/distilbert-base-uncased

Number of total training steps is decided from 500 to 2000. In Adult-Confounded and CMNIST, full-batch training is implemented due to enough memory space. In Waterbirds and SNLI, batch size is chosen among 128, 256 and 512. The choices of learning rate and optimizer depend on the dataset. For Adult-Confounded, CMNIST and Waterbirds, a learning rate between 2e-4 and 2e-3 is considered. We take Adam as the optimizer for Adult-Confounded and CMNIST, and choose SGD for Waterbirds. For SNLI, a smaller learning rate in {2e-5, 3e-5, 5e-5, 1e-4} is selected when fine-tuning DistilBERT with AdamW.

### B.3 Oracle Settings on Adult-Confounded

Given a biased training set and two sensitive features, *race* and *sex*, $\mathcal{E}_{\text{oracle}}$ for IRM is constructed according to Table 8. The correlations between variant features and target are maximized within each environment and diversified across environments. As implied by [5, 22], spurious correlations are supposed to be eliminated when an invariant learning algorithm converges properly.

Table 8: Oracle environments for IRM on Adult-Confounded, where $e_i \in \text{supp}(\mathcal{E}_{\text{oracle}})$ represents the $i$-th environment.

|       | $Y = 1$ | | $Y = 0$ | |
|-------|-----------|--------|-----------|--------|
|       | Race | Sex | Race | Sex |
| $e_1$ | Non-black | Male | Black | Female |
| $e_2$ | Non-black | Female | Black | Male |
| $e_3$ | Black | Male | Non-black | Female |
| $e_4$ | Black | Female | Non-black | Male |

### B.4 Biased Model for SNLI

To define subsets for evaluation on SNLI, we follow the labeling procedure in [9]. Specifically, k-fold cross validation ($k = 5$) is applied on the training set. We fine-tune BERT [7] with hypothesis as its only inputs on $k - 1$ folds, and score the left-out $k$-th set. For the development and testing sets, we score each example with average predictions from $k$ different models. The accuracy of the biased model is approximately 68%. Finally, we set two thresholds $(t_1, t_2)$, defined by [9], to (0.2, 0.5), where $t_1$ is used to identify unbiased data, and $t_2$ is used to define bias aligned and misaligned sets.

## C Additional Empirical Results

### C.1 Synthetic Data for Regression Problem

We further validate our work on regression problem with a synthetic dataset proposed in [23]. The features are $X = H[X_c, X_v] \in \mathbb{R}^d$, and the target is generated by $Y = f(X_c) + \epsilon$, where $H$ is an random orthogonal matrix for scrambling features and $f(\cdot)$ is a non-linear function. $X_c$ are invariant features that $P(Y|X_c)$ is consistent across environments, while $X_v$ are variant features that $P(Y|X_v)$ can arbitrary change according to the following data sampling mechanism:

$$\hat{P}(x_i, y_i) = |r|^{(-5 \cdot |y_i - sign(r) \cdot X_v^*|)} \tag{14}$$

In Equation 14 where $|r| > 1$, $r$ controls the spurious correlation between the certain variable $X_v^* \in X_v$ and the target label $Y$. Specifically, larger $|r|$ represents stronger correlation, and the sign of $r$ indicates the direction of correlation. In the training set, there are 1000 examples generated from the environment with $r = 2.3$ and 100 examples from that with $r = -1.1$. The environment labels are unavailable as in the previous experiments. For testing, two scenarios are considered. First, we define two environments, IID and OOD, to evaluate the generalization under dramatic distributional shifts. In IID, 1100 examples are sampled with the same procedures as training data. In OOD, 1000 examples are generated from the environment with $r = -2.5$. Secondly, following [23], we evaluate the stability over six testing environments where $r \in \{-2.9, -2.7, -2.5, ..., -1.9\}$.

In our regression task, the evaluation metric is mean square error. For all methods, MLP with one hidden layer of 1024 neurons is utilized. When calculating the label independency term $L_{\text{LI}}$ for

EDNIL, we discretize $Y$ by quartiles. Empirically, such efficient estimation can improve the quality of environment inference to some degree. We leave more precise approximations of mutual information between discrete and continuous variables for future work.

**Results** The results of the first testing scenario are listed in Table 9. Among all methods, EDNIL obtains the most consistent scores across IID and OOD. The performance degradation when $\beta = 0$ suggests the importance of $L_{\mathrm{LI}}$.

Table 10 shows the results of the second scenario. As in [23], *Mean Error* and *Std Error* represent the mean and standard deviation of errors over six testing environments respectively, both of which are averaged over 20 runs. Similar to the first scenario, EDNIL performs the best and most robustly in out-of-distribution settings. The estimated $L_{\mathrm{LI}}$ also gains empirical improvements in this test.



Table 9: Testing mean square errors on synthetic regression dataset (Scenario 1).

|  | IID | **OOD** |
|---|---|---|
| ERM | **0.772** $\pm$ **0.079** | 5.431 $\pm$ 0.461 |
| EIIL | 1.629 $\pm$ 0.174 | 3.675 $\pm$ 0.756 |
| KerHRM | 1.246 $\pm$ 0.339 | 3.612 $\pm$ 1.082 |
| EDNIL | 1.971 $\pm$ 0.183 | **2.253** $\pm$ **0.422** |
| EDNIL$_{\beta=0}$ | 1.733 $\pm$ 0.340 | 2.933 $\pm$ 0.808 |

Table 10: Mean square errors of stability test on synthetic regression dataset (Scenario 2).

|  | Mean Error | Std Error |
|---|---|---|
| ERM | 5.367 | 0.217 |
| EIIL | 3.623 | 0.188 |
| KerHRM | 3.526 | 0.151 |
| EDNIL | **2.218** | **0.103** |
| EDNIL$_{\beta=0}$ | 2.879 | 0.151 |



## C.2 CMNIST with Different Color Noises

In this task, the learning effects of all methods are tested under different strengths of spurious correlation at train time. We select CMNIST with fixed label noise 0.2, and adjust overall color noise $e$ from 0.1 to 0.3 to generate five different training sets. For testing, $e = 0.1$ and $e = 0.9$ are considered. Given color label $C$ and target $Y$, samples with $C = Y$ are more than those with $C \neq Y$ when $e = 0.1$, where the direction of spurious correlation is aligned with that in the training sets. On the other hand, when $e = 0.9$, samples with $C \neq Y$ are in the majority. Due to the reversed correlation, models relying on the variant feature (i.e. color $C$) are vulnerable to this setting.

**Results** The results are plotted in Figure 4. As the training color noise increases, the generalization of ERM improves since the spurious correlation decreases at train time. Meanwhile, as indicated in [6], EIIL fails because the reference model, i.e. ERM, is no longer a pure variant predictor. For KerHRM, the instability of inferred environments results in large standard deviation, especially when training color noise is small. In comparison, given different strengths of spurious correlation for training, EDNIL can distinguish invariant features from variant ones, and performs more consistently as IRM (with $\mathcal{E}_{\mathrm{oracle}}$) does.


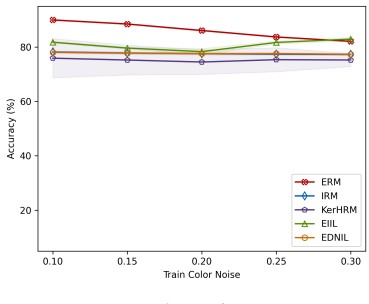

(a) Test color noise $e = 0.1$

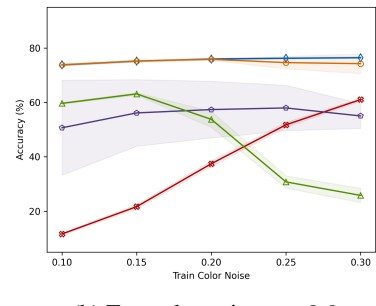

(b) Test color noise $e = 0.9$


Figure 4: Testing accuracy (%) of CMNIST. Testing set with color noise $e = 0.9$ is more challenging since the direction of spurious correlation is opposite to that in training.