# OpenReview forum: "Environment Diversification with Multi-head Neural Network for Invariant Learning"
_NeurIPS.cc/2022/Conference — NeurIPS 2022 Accept_

### Official Review · Reviewer_QRiV · 2022-07-10

**Rating:** 7
**Confidence:** 3
**Soundness:** 4 excellent
**Presentation:** 3 good
**Contribution:** 4 excellent

**Summary:**

In the present paper, the authors tackle the challenging problem of learning a supervised ML model capable of identifying and adapting to different environments hidden in the training data. The method, EDNIL, is composed of two jointly learned models, that take care of the environment identification, the learning of the invariant representations and the label predictions, produced by a multi-headed neural network. The proposed model is compared to different alternative models from the literature of the field, in different challenging benchmarks, and the results show that it closely achieves the best possible invariant learning performance.

AFTER REBUTTAL
The authors have addressed the main concerns raised during the review. I confirm my assessment of this paper as solid and deserving of acceptance in the conference.

**Questions:**

- In general, the presence of way too many acronyms can confuse the reader. I do understand that presenting all this material in 9 pages is quite a challenge but the density is a bit excessive in some passages of the manuscript.
- lines 74-93 seem a bit hard to follow, especially about the role of Phi and Xv.
- the role of beta and gamma seems important, but the authors only give space in the main text to the discussion of the extreme cases beta=0 and gamma=0. Can a few comments be added about the range of optimal values for these parameters? How hard is it to fine-tune them and how variable are they across different learning problems?
- lines 166-169 (and later figure 7 and its description): what is the intuitive explanation of the reduced dependence on the training stage for the ERM initialization. Is it about the role of the L_IP term, that avoids trivial environment split scenarios?
- line 195 is not clear, probably just needs rewriting.
- lines 227-230 the sentence seems to be missing a verb and becomes a bit obscure.
- line 236: removing the coefficient beta -> in what sense? setting it to 0, or to 1? Probably rephrasing could help.
- In general, it seems to me that this work could be very impactful also for the field of ML fairness, where the different environments could represent protected categories that are misrepresented in the available data and the learning model risks to become biased. Did the authors consider this connection?


**Limitations:**

I think the main limitations of the paper are well acknowledged by the authors.

**Strengths And Weaknesses:**

The authors introduce a very flexible and effective learning framework, incorporating the main strengths of the previously introduced models and surpassing many of the limitations thereof. The novel method is shown to perform extremely well when compared to its competitors, achieving close to the performance of IRM with access to the oracle environments.

I think the main weakness of this paper is in the density of the presentation and in the abundance of acronyms that make the read quite challenging, especially for a non-expert.

---

> ### Author Response · Authors · 2022-08-02
> **Response to Reviewer QRiV**
>
> We thank the reviewer for the comments and feedback. Below we discuss the questions the reviewer raised:
>
> **Q1. *"... too many acronyms can confuse the reader ..."***
>
> **Response:** We appreciate the suggestion. In the revised paper, we have reorganized and simplified some paragraphs in Section 2 and 3 for improving readability. In Section 2, line 74-93 in particular, we have replaced the mathematical definitions with brief and intuitive explanations. In Section 3, as recommended by Reviewer T3xK, we have added the symbols alongside the loss names. We have also combined the introduction of objectives with loss functions for better readability.
>
> **Q2. *"Lines 74-93 ..."***
>
> **Response:** In the newly uploaded paper, we have revised line 74-93 to provide more intuitive explanations of $\Phi$, $X_c$ and $X_v$. The revised paragraph is expected to convey that input features $X$ are assumed to be divided into invariant features $X_c$ and variant features $X_v$, where $X_v$ are not uniformly informative for prediction across environments, and thus we attempt to find an encoder $\Phi$  for obtaining invariant features $X_c \cong Φ(X)$.
>
>
> **Q3. *"The role of beta and gamma seems important ..."***
>
> **Response:** The optimal values of $\beta$ and $\gamma$ depend on the training datasets, choice of encoders, as well as the choice of other hyper-parameters. Nevertheless, we found that it is not difficult to find optimal values via in-distribution validation data. Ranges of values of $\beta$ and $\gamma$ are suggested in supplementary file Section B.2. For the classification problems, the final values of $\beta$ and $\gamma$ are between 2 to 10. For the regression problem mentioned in supplementary file Section C.1, the values are usually less than 1. The actual values for each problem are provided in the source code.
>
> **Q4. *"...the reduced dependence on the training stage for the ERM initialization ..."***
>
> **Response:** Unlike EIIL, who directly takes ERM as reference model heavily relying on variant features, EDNIL regards ERM initialization only as the role in facilitating feature extraction. The identification of variant features is mainly learned from the graphical model (Figure 2) and the designed loss functions (Formula 6-9). As Condition 3 and 4 are satisfied, $M_\text{EI}$ can obtain useful environments for invariant learning. In conclusion, due to the more delicate loss functions probing variant features in EDNIL, the quality of environment inference depends less on ERM initialization.
>
> **Q5. *"line 195  ..."***
>
> **Response:** Thank you for the reminder. We have rewritten the sentence in our revised paper.
>
> **Q6. *"lines 227-230 ..."***
>
> **Response:** Thank you for the reminder. We have rewritten the sentence in our revised paper.
>
> **Q7. *"line 236 ..."***
>
> **Response:** Thank you for the reminder. We change the word “removing” to “setting it to zero” for clarity in the revised paper.
>
> **Q8. *"... very impactful also for the field of ML fairness ..."***
>
> **Response:** Yes, we have acknowledged this connection and discussed our possible impacts on ML algorithmic fairness in our supplementary file Section D. More details about the connections between invariant learning and algorithmic fairness are elaborated by Creager et al. [1]. In fact, our superiority performance on Adult-Confounded dataset is one of the evidence that EDNIL has potentials for learning a fair predictor without discrimination against sensitive demographic groups. We anticipate that one can extend our work to more fairness benchmarks and metrics in the future.
>
> [1] Creager, Elliot et al. “Environment Inference for Invariant Learning.” ICML (2021).

---

### Official Review · Reviewer_T3xK · 2022-07-11

**Rating:** 7
**Confidence:** 5
**Soundness:** 3 good
**Presentation:** 2 fair
**Contribution:** 4 excellent

**Summary:**

The authors propose a novel environment-free invariant learning method that uses an auxiliary network to learn environment-specific features, from which environment inferences can be derived. These inferred environments labels are then used to train and invariant model.

UPDATE 2022-aug-05: The paper's clarity has been improved so I am increasing my score by one.

**Questions:**

* I was a little unclear on the motivation for using the environment (in)dependency objective - I can see its utility in the ablation studies, but what was the reason, conceptually speaking, to introduce this in the first place?
* Beyond looking at the value of L_ED during training, can an experiment be formulated to test whether the inferred environments (samples from P(e|X,Y)) are indeed “diverse” in the way the authors are hoping for?
* Section 3 is a bit hard to parse given that the three objectives are introduced in several places (lines 129–133, Fig 1, Eqn 6). I would suggest that the authors take care to discuss these in the same order each time and include the symbols alongside the objective names (e.g. “Label dependency objective” => “Label dependency objective ($L_{LD}$” in line 129). I also found it a bit confusing that some objectives are for the inference net Phi while others are for the invariant learner. While I can see that Fig 1 attempts to make this distinction by placing the loss terms in different parts of the figure, it would be even clearer if the networks (or their parameters) were used as arguments to the losses, e.g. L_ED(Phi, Psi) = L_ED(Psi) + beta L_LD(Psi) + gamma L_IP(Phi) in Eqn 6). I would also order the terms in Eqn 6 in the same order as they appear in the text.
* Since the variance of per-env losses is used as L_IP, it is worth mentioning REx (https://arxiv.org/abs/2003.00688) as a related work, since it uses a similar objective (with known environments).
* Regarding the tabular experiments, there are some limitations of the UCI-Adult dataset (https://openreview.net/forum?id=bYi_2708mKK) that would be relevant/interesting to NeurIPS readers/attendees (perhaps in a footnote).
* Environment-free invariant learning is a very ambitious problem, which could be difficult to solve in some settings. I would be curious to hear about experimental settings where EDNIL fails; I think these would interest the reader and also make the limitations of this method a bit more clear.
* When mentioning how the training/validation splits for Waterbirds are merged [lines 290–291], it would be good to mention that the original validation split contains all four groups in equal proportions, whereas original training split has unequal proportions.
* I find the double negatives needed to parse Table 1 to be slightly confusing.


**Limitations:**

No [line 430]. I suggest that the authors add a discussion of possible societal impacts to a future revision. Given the connections of invariant learning to fairness (https://arxiv.org/abs/2010.07249) and the use of a fairness benchmark in the experiments, there should be plenty to discuss.

**Strengths And Weaknesses:**

I like the idea of the paper, and was impressed by the experimental results showing that the proposed method EDNIL realizes good worst-group performance in a variety of settings, while addressing the sensitivity to initialization that we see in EIIL. In my opinion, the experimental results are sufficient but the presentation should be improved (see below for specific suggestions).

Strengths
* I like the experimental results. The “effortless initialization” result (Fig 7), showing that the proposed method is not sensitive to initial ERM features, which is a known issue with EIIL, is especially impressive.
* The paper does a good job of describing related approaches to environment-free invariant learning, and comparing against these methods empirically.
Weaknesses
* The paper can be hard to read at times. Some parts would benefit from additional exposition. For example the idea of “diversity” of inferred environments is mentioned in passing with a citation, but it is not fully explained what type of diversity is needed, and how it can be measured formally? The cited paper (https://arxiv.org/abs/2004.05007) mentions that the correlation pattern between Y and spurious features should vary across environments…is the outcome we hope for when introducing L_ED?

---

> ### Author Response · Authors · 2022-08-02
> **Response to Reviewer T3xK (Reference)**
>
> [1] Hashimoto, Tatsunori B. et al. “Fairness Without Demographics in Repeated Loss Minimization.” ICML (2018).
>
> [2] Lahoti, Preethi et al. “Fairness without Demographics through Adversarially Reweighted Learning.” ArXiv abs/2006.13114 (2020).
>
> [3] Yan, Shen et al. “Fair Class Balancing: Enhancing Model Fairness without Observing Sensitive Attributes.” Proceedings of the 29th ACM International Conference on Information & Knowledge Management (2020).

---

> ### Author Response · Authors · 2022-08-02
> **Response to Reviewer T3xK**
>
> We thank the reviewer for the comments and feedback. Below we discuss the questions the reviewer raised:
>
> **Q1. *"... the motivation for using the environment (in)dependency objective ..."***
>
> **Response:** As discussed in Section 3.1.2 (Line 159-161), with d-separation, $\mathcal{E}_\text{learn}$ and $Y$ are independent in the graphical model (Figure 2), a sufficient condition of Condition 3 and 4. As a result, we restrict their dependency via $L_\text{LD}$. In practice, optimizing $M_\text{EI}$ with $L_\text{LD}$ can prevent a trivial solution that environment partitions are purely determined by labels $Y$, which is undesirable for invariant learning as shown in the ablation study.
>
> As suggested by the reviewers, in our revision, the introduction of learning objectives has been combined with loss terms, i.e. $L_\text{ED}$, $L_\text{LD}$ and $L_\text{IP}$, in order to improve the readability. That is, label dependency objective $L_\text{LD}$, which acts as a regularization term in $L_\text{EI}$, will not be introduced in the first place.
>
>
> **Q2. *"... test whether the inferred environments ... are indeed “diverse” ..."***
>
> **Response:** Yes. As shown in Figure 6, we have conducted a visualization experiment after the models are trained and values of hyper-parameters are determined. Given the variant feature (i.e. color) in CMNIST, the correlations with $Y$ in $\mathcal{E}_\text{learn}$, inferred from EDNIL, almost resemble the ideal diversity (i.e. $\mathcal{E}_\text{oracle}$). As such, the effectiveness of our designed objectives can be empirically proven.
>
> **Q3. *"Section 3 is a bit hard to parse ..."***
>
> **Response:** Thank you for the comments. We have reorganized our Section 3 for better readability. In particular, we combine the introduction of objectives with loss functions, add the symbols alongside the loss names, and elaborate that $L_\text{EI}$ is for updating $M_\text{EI}$ (including $\Psi$ and $f^e$), and $L_\text{IL}$ is for updating $M_\text{IL}$ (including $\Phi$).
>
>
> **Q4. *"... worth mentioning REx ..."***
>
> **Response:** Thank you for the useful suggestion. We have cited this related work in our revision.
>
> **Q5. *"... some limitations of the UCI-Adult ..."***
>
> **Response:** In our revised paper, we have put some discussions about it in a footnote (Page 6). Thank you.
>
> **Q6. *"... experimental settings where EDNIL fails ..."***
>
> **Response:** We have realized that the task can be very challenging when biases are introduced in the meta level. For example, in Waterbirds and SNLI, the performance gaps between EDNIL and IRM (assuming oracle environments are given) are more significant than those in Adult-Confounded and CMNIST.  A possible reason is that it is hard to distinguish variant and invariant features according to raw features. Therefore, representation learning plays an important role in extracting informative features. Although we have shown the effectiveness and superiority of EDNIL when incorporating representation learning into environment-free invariant learning, more investigation can be conducted to achieve better performance in the future.
>
>
> **Q7. *"... the training/validation splits for Waterbirds are merged ..."***
>
> **Response:** Thank you for the suggestion. In the revised paper, we have mentioned the information of original splits in a footnote (Page 8).
>
>
> **Q8. *"... the double negatives ..."***
>
> **Response:** We are sorry for the confusing presentation. In the revised paper, we have replaced “No need of prior knowledge of environments”, “No need of delicate initialization”, and “No scalability issue” with “Unsupervised”, “Insensitive initialization”, and ”Efficiency”.
>
>
> **Weakness (Diversity of environments)**
>
> **Response:** Thank you for the advice. We have discussed more about the conditions and diversity of environments in Section 2, and explained that the design of $L_\text{ED}$ can realize the insight of capturing diverse variant relationships to enlarge the discrepancy of spurious correlations between environments in Section 3.
>
> For the measurement of diversity, as discussed in Q2, targeted bias in each environment can be visualized to verify and explain the effectiveness of our method (Figure 6) at test time.
>
>
> **Limitations (Connection to fairness)**
>
> **Response:** We appreciate the suggestion. As shown in the supplementary file Section D, we have acknowledged and discussed the potential impacts on ML algorithmic fairness. In particular, our superiority on Adult-Confounded is one of the evidence that EDNIL has potentials to learn a fair predictor without discrimination against sensitive demographic groups. In the revised supplementary file (Section D), we have added more related works [1, 2, 3] to strengthen the connection to fairness problems.

---

> > ### Comment · Reviewer_T3xK · 2022-08-05
> > **improved clarify**
> >
> > Thanks to the authors for their considered response.
> >
> > Taking a quick look at the updated pdf, and reading the response, it does seem that the clarity of the submission has been improved. While there may still be room for improvement on this front, I am planning to increase my score by one.
> >
> > Regarding societal impacts, I'm not sure I agree with the authors that the environment-free invariant learning would have no negative societal impact [paraphrasing Appendix D]. This is just one example, and it is a subtle point, but consider the paper "AI recognition of patient race in medical imaging: a modelling study" by Gichoya et al, which studies whether AI systems can infer patient race from medical images alone (which, in principle, shouldn't be possible). When the study was disseminated in the news, the result that AI systems succeeded at this prediction task was (erroneously) interpreted by some as an indication that racial attributes are essential to a person (as opposed to socially ascribed). What happens if categories like race are automatically inferred by a method like this?
> >
> > According to me, it is perfectly appropriate to be self-critical when evaluating potential societal impacts. And I don't see any reason to reject this paper on ethical grounds. So I would encourage the authors to think critically about these matters and be candid with the reader, rather than painting an overly rosy picture of things.
> >
> > NOTE: the author checklist question 1c can now be updated from 'No' to 'Yes'

---

> > > ### Author Response · Authors · 2022-08-07
> > > **Response to feedback**
> > >
> > > We thank reviewer T3xK for the feedback and insights. We agree that misinterpretation of machine outputs needs to be considered as a potential cause of negative societal impacts. To address this issue, we have revised the supplementary file again to include relevant discussions in Section D. In particular, we suggest that adequate and careful preparations and analyses are required before drawing conclusions from the inferred invariant relationships.
> > > > *Last but not least, it is worth mentioning some cautions. Since the invariant learning algorithm claims to find invariant relationships, one might cast more attention on feature importance of the invariant model and even incorporate the results into further research or applications. However, the results are reliable only when the model is trained appropriately. Insufficient data collection or careless training process, for example, can certainly affect the identification of invariant features, and thus mislead experimental findings. As a result, we believe that adequate and careful preparations and analyses are essential before drawing conclusions from the inferred the invariant relationships.*

---

> > > > ### Comment · Reviewer_T3xK · 2022-08-09
> > > > **societal impacts**
> > > >
> > > > Thanks - this feels like a more well-rounded societal impact discussion now.

---

### Official Review · Reviewer_SNzy · 2022-07-12

**Rating:** 6
**Confidence:** 4
**Soundness:** 3 good
**Presentation:** 2 fair
**Contribution:** 2 fair

**Summary:**

Paper proposes a new method to infer environment labels that can then be used for invariant learning (such as IRM methods). Essentially, it uses a neural network for clustering data into environments with appropriate loss functions. The method is more scalable and less sensitive to initializations than the prior methods. Experiments show significant improvements for distribution-shifted tasks without environment labels.

**Questions:**

1. Problem 2.1 needs to be explained more, discussing the use of Shannon entropy. Also clarify the novelty; I believe Equation (3) was proposed in [1] but Equation (4) is a contribution?
2. In Equation (4), is it enough that LHS is just > 0 while being arbitrarily small? Should the LHS be maximized instead?
3. The proposed method in Equation (5) needs more explanation on why it was chosen to be this way. I believe it is done to ensure that data is divided into environments that maximize label prediction in the individual environments. This is interesting but requires more justification in the text. Related to my previous comment, $H(Y | X_v) - H(Y | X_v, \mathcal{E}_\text{learn})$ might be actually maximized with this objective, and not just >0.

4. Discussing $L_{IP}$ before discussing the $M_{IL}$ model hampers clarity. It should also be clarified/reiterated that $L_{IP}$ is only used to update the $M_{EI}$ network and not $\Phi$, and that there are two separate invariance preserving objectives (one for $M_{IL}$ and one for $M_{EI}$).
5. It is very interesting that the number of environments (clusters) need not be specified accurately for EDNIL to work (as seen in Figure 6). More discussion regarding why this is so will be helpful, i.e., which part of the loss ensures this.


Minor:

1. Please use \text{} in math equations for non-variables, such as \text{learn}, \text{LD}, etc. Also, use \exp in Equation (5).
2. Line 19: “… aims at learning causality expected to be robust …” should be “..aims at learning causal features expected to be robust..”.
3. Line 106: should be “HRM equips a joint learning framework”.

[1] Jiashuo Liu, Zheyuan Hu, Peng Cui, Bo Li, and Zheyan Shen. Heterogeneous risk minimization. In International Conference on Machine Learning (ICML), 2021.

**Limitations:**

Yes

**Strengths And Weaknesses:**

Strengths:

1. Paper tackles an important challenge of robust learning with no environment labels and solves the problems with existing works.

2. Experiments in varied tasks, including images and text, show significant performance improvements. Experiments also help answer questions about the effect of initializations, effect of number of environments.


Weaknesses:

1. Clarity can be improved a lot; especially Section 2.2 and Section 3 (Methodology). As a result, the paper’s contributions are muddled with existing work (for example, which of the learning objectives and loss terms are novel?).
2. Paper can be improved a lot with better justifications for the choices made in the paper. Currently the loss functions seem a little arbitrary; for example, why are Equations (5) and (9) the best ways to achieve their respective goals.
3. There are no theoretical guarantees provided by the prior works, but this can be future work.

---

> ### Author Response · Authors · 2022-08-02
> **Response to Reviewer SNzy (Reference)**
>
> [1] Lin, Yong et al. “ZIN: When and How to Learn Invariance by Environment Inference?” ArXiv abs/2203.05818 (2022).
>
> [2] Liu, Jiashuo et al. “Heterogeneous Risk Minimization.” ICML (2021).
>
> [3] Liu, Jiashuo et al. “Kernelized Heterogeneous Risk Minimization.” NeurIPS (2021).
>
> [4] Chen, Guanzhou et al. “Training Small Networks for Scene Classification of Remote Sensing Images via Knowledge Distillation.” Remote. Sens. 10 (2018): 719.
>
> [5] Jang, Eric et al. “Categorical Reparameterization with Gumbel-Softmax.” ArXiv abs/1611.01144 (2017).
>
> [6] Krueger, David et al. “Out-of-Distribution Generalization via Risk Extrapolation (REx).” ICML (2021).

---

> ### Author Response · Authors · 2022-08-02
> **Response to Reviewer SNzy**
>
> We thank the reviewer for the comments and feedback. Below we discuss the questions the reviewer raised:
>
> **Q1. *"Problem 2.1 needs to be explained more ..."***
>
> **Response:** Equation 3 and 4 are both from ZIN [1] while Equation 3 indeed was discussed earlier in the HRM paper [2]. We note that our previous presentation sent a misleading message that we propose to optimize Equation 3 and 4 directly. Instead, the contribution of our work is the proposed multi-head neural network with carefully designed loss functions. We propose an objective containing $L_\text{IP}$, $L_\text{ED}$ and $L_\text{LD}$. In particular, we show that minimization of $L_\text{IP}$ and $L_\text{ED}$ (maximizing the LHS of Equation 4) are sufficient conditions for Equation 3 and 4. The other loss function, $L_\text{LD}$, comes from the proposed graphical model to avoid trivial environment assignment.
>
> In the revised paper, we explicitly introduce Equation 3 and 4 as the conditions proposed by Lin et al. [1], and move our novel learning objectives / loss terms to Section 3 for distinction.
>
>
> **Q2. *"In Equation (4), is it enough that LHS is just > 0 ... maximized instead?"***
>
> **Response:** In theory, LHS of Equation 4 can be larger than zero but arbitrarily small. As long as the invariant learning algorithm for $M_\text{IL}$ is powerful enough, variant features will still be excluded for prediction. However, as implied by [2, 3], capturing more diverse spurious relationships in environments can reduce the difficulty in identifying variant and invariant features in practice. Therefore, we choose to maximize the LHS to meet both theoretical (Equation 4) and empirical requirements.
>
> We would also like to clarify that Equation 4 is not the learning objective but a theoretical condition concerning dependency. As our previous presentation was misleading, we separate the discussions of conditions and objectives into Section 2 and 3 in the revised paper.
>
>
> **Q3. *"... Equation (5) needs more explanation ..."***
>
> **Response:** The design of Equation 5 corresponds to the graphical model (Figure 2). Particularly, the environment labels are associated with the relationship between variant features and target. Inspired by classical multi-class classification problems, we resort to a softmax function with input of negative $l(f^e(\Psi(X)), Y)$ for the probabilities of assignments. For training, $L_\text{ED}$ is minimized to diversify the correlations between environments. As discussed in Q2, the LHS of Equation 4 is supposed to be maximized, resulting in both theoretically and empirically effective environments for invariant learning algorithms.
>
> We have explained the motivations and explanations of Equation 5 and $L_\text{ED}$ in the revised paper (Line 139-157). Thank you.
>
>
> **Q4. *"Discussing $L_\text{IP}$ ..."***
>
> **Response:** Agree. We have clarified the purpose of $L_\text{IP}$ in Section 3.1, and distinguished it from the learning of $M_\text{IL}$ (i.e. $L_\text{IL}$ in Section 3.2) in our revision.
>
>
> **Q5. *"... the number of environments ..."***
>
> **Response:** As shown in Figure 6, EDNIL can control the sparsity of environments, resulting in the insensitivity to the specified number of environments. We attribute this to the usage of softmax function with temperature $\tau$ (Equation 5). Several studies [4, 5] demonstrate that smaller temperature in softmax function will lead to more concentrated output distribution. In our case, as described in supplementary file Section B.2, the recommended range of temperature $\tau$ is from 0.1 to 0.5, which is relatively small. When the number of environments is larger than the oracle one, only informative environments are inferred as a result of concentration.
>
> **Weakness 1 (Clarity in Section 2.2 and 3)**
>
> **Response:** We appreciate the helpful suggestions. We have revised the paper accordingly.  As the response in Question 1, we explicitly introduce Equation 3 and 4 as the conditions proposed by Lin et al. [1], and move our proposed learning objectives / loss terms to Section 3 for clarification.
>
> **Weakness 2 (Justifications for the choices)**
>
> **Response:** Thank you for the advice. We add more justifications about the choices of loss functions in our revised paper. The main motivations and reasons for using Equation 5 and 8 (or 9 before revision) are as follows:
> 1. For Equation 5, the explanations are covered by Question 2 and 3.
> 2. For $L_\text{IP}$ (Equation 8), it is designed as the contrary of $L_\text{ED}$ (Equation 6) to prevent the inferred environments from diversifying the risks of the learned invariant model. As the variance of environment risks is regularized, the existing invariant relationship will remain consistent across the environments to satisfy Condition 3. Similar idea of using the variance can be found in [6] (but solving different task).
>
> **Minor 1 to 3**
>
> **Response:** All minors are corrected in the revised paper.

---

> > ### Comment · Reviewer_SNzy · 2022-08-08
> > **Thanks for the clarifications**
> >
> > Thanks for addressing all my concerns. I will update my score soon (currently I am unable to access the revised paper).

---

> > > ### Comment · Reviewer_SNzy · 2022-08-09
> > > **Clarity has improved but one question.**
> > >
> > > Looking at the revised paper, it is not immediately clear how Equation (6) induces maximization of Condition 4. I agree that $H(Y | X_v, E_\text{learn})$ is minimized but shouldn't $H(Y | X_v)$ be maximized as well?

---

> > > > ### Author Response · Authors · 2022-08-09
> > > > **About Equation 6**
> > > >
> > > > We thank Reviewer SNzy for reading our revised paper.
> > > >
> > > > We would like to clarify that Equation 6 indeed aims at maximizing $H(Y | X_v) - H(Y | X_v, \mathcal{E}_\text{learn})$ instead of minimizing $H(Y | X_v, \mathcal{E}_\text{learn})$ only. Specifically, the difference can be represented by conditional mutual information; namely, $I(Y; \mathcal{E}_\text{learn} | X_v) = H(Y | X_v) - H(Y | X_v, \mathcal{E}_\text{learn})$. We argue that $I(Y; \mathcal{E}_\text{learn} | X_v)$ can be optimized by $L_\text{ED}$, as the dependency between $Y$ and $\mathcal{E}_\text{learn}$ is expected to be maximized given variant representations.

---

### Official Review · Reviewer_8Cwm · 2022-07-13

**Rating:** 5
**Confidence:** 4
**Soundness:** 2 fair
**Presentation:** 2 fair
**Contribution:** 2 fair

**Summary:**


The paper discusses the problem of shift between train and test data distributions. The authors assume that training data originate from multiple 'environments', each with particular distribution. They propose a new framework (EDNIL) through which they can infer the environments from data (without human labeling) and then train an environment invariant model. The principal assumption is that of the existence of `invariant features` - the distribution of the outputs conditioned on these is the same across all the environments. The loss is a combination of multiple terms motivated through information theoric arguments. The algorithm consists of alternating steps to infer the environments and train the invariant model. Multiple experiments are provided to corroborate the effectiveness of the method.

**Questions:**

1) While the intro speaks about a shift between train and test distributions, in the end it seems you address the problem of a mix of multiple distributions in both train and test. That is the train and test are not fundamentally different, they are just both formed through a combination of multiple distributions, is this right?
2) If so, what is the meaning of $\mathcal{E}_{all}$ in your setup?
3) Do I assume correctly that the output and input spaces are aligned across the environments (including the supports of the distributions)?
4) What are $\Phi$ and $\Psi$? Are these some general functions such as small MLPs?
5) In equation (5) softmax you sum across the E_learn invoronments, Does it mean that the number of environments needs to be fixed beforehand?
6) How is the environment decided at test time, when no labels are available? Which head shall be used?

**Limitations:**

Societal impact - not addressed, not directly relevant.
Limitations - invalidity of graphical model assumption is mentioned. Yes, is reasonable.

**Strengths And Weaknesses:**

Train/test distribution shift is an important and fully answered problem, particularly relevant for all practical applications where the standard iid assumptions typically do not hold. As such the paper addresses a significant question for the community.
I found the paper somewhat difficult to parse and follow. Most importantly, I found the description of the whole setup (shift from train to test distributions or a mix of multiple environments in train and test) rather confusing. But I have more question marks (see Questions) indicating a fairly low clarity of the paper.
In the end the solution is an alternating optimization in which through the first step the environments are inferred though a form of supervised clustering (similarity metric is the model accuracy) and then this info is used to train a prediction model. If my interpretation is correct, this does not seem very novel.

---

> ### Author Response · Authors · 2022-08-02
> **Response to Reviewer 8Cwm**
>
> We thank the reviewer for the comments and feedback. Below we discuss the questions the reviewer raised:
>
> **Weakness (Novelty of our work)**
>
> **Response:** Different from the SOTA models who model the environment inference task as adversarial learning (i.e. EIIL) or clustering task (KerHRM), our solution models it as a classifical multi-class learning task with carefully-crafted loss functions. We propose an objective containing $L_\text{ED}$, $L_\text{LD}$ and $L_\text{IP}$. In particular, we show that minimization of $L_\text{IP}$ and $L_\text{ED}$ are sufficient conditions for Equation 3 and 4. The other loss function, $L_\text{LD}$, comes from the proposed graphical model to avoid trivial environment assignment.
>
> Our solution has several advantages: 1. The objective function attempts to achieve the ideal conditions proposed by [5] 2.  Different from EIIL, our model does not heavily rely on initialization. 3. Different from HRM and KerHRM, our model can be efficiently trained with deep neural networks.
>
> Note that we do not consider alternative training as the main novelty of this work.
>
>
> **Q1. *"... the problem of a mix of multiple distributions in both train and test ..."***
>
> **Response:** The test distribution is not assumed to be a mixture of multiple distributions or environments.
>
> Specifically, our approach belongs to invariant learning, where the goal is not to find a predictor via interpolation (i.e., fitting a mixture of known distributions) but extrapolation (i.e., finding a predictor optimal to any unseen distributions given some observed distributions). It is expected to be achieved by the IRM-based objective (Equation 10) with appropriate environments (inferred via our method).
>
> Take CMNIST proposed by IRM [1] for example. The data generation process is $X_c \rightarrow Y \rightarrow X_v$, where $X_c$ is the shape of digit and $X_v$ is the color of digit. Across $\rm{supp}(\mathcal{E}_\text{all})$, $X_c \rightarrow Y$ is invariant, but $Y \rightarrow X_v$ can be arbitrarily. In particular, color of each digit (i.e. $X_v$) is determined by flipping $Y$ with probability $p$. Each environment $e \in \rm{supp}(\mathcal{E}_\text{all})$ chooses an arbitrary $p$ ranging from $0$ to $1$. The training data are collected from $\rm{supp}(\mathcal{E}_\text{tr})$ consisting of two environments with $p = 0.1$ and $0.2$ (i.e. the mixture of environments / distributions). As for testing, an extreme environment with $p = 0.9$, which belongs to $\rm{supp}(\mathcal{E}_\text{all}) \ \backslash \ \rm{supp}(\mathcal{E}_\text{tr})$, is considered to evaluate the performance of generalization.
>
> **Q2. *"...what is the meaning of $\mathcal{E}_\text{all}$ in your setup?"***
>
> **Response:** Following [1, 2, 3, 4], $\mathcal{E}_\text{all}$ contains all possible environments / distributions satisfying certain data generation process. As the response in Question 1, we intend to find a predictor optimal to unseen environments given some observed environments.
>
> **Q3. *"... the output and input spaces are aligned across the environments ..."***
>
> **Response:** Regarding the distributions $P^e (X^e,Y^e)$ where $e \in \rm{supp}(\mathcal{E}_{all})$, we do assume the output and input spaces are aligned, while we do not assume the supports of the distributions are also aligned.
>
> **Q4. *"What are $\Phi$ and $\Psi$ ..."***
>
> **Response:** Φ and Ψ are encoders transforming data into invariant and variant representations respectively. We assume Φ and Ψ can be high or low-capacity neural networks. In our experiments, Φ and Ψ are MLPs for CMNIST and Adult-Confounded, Resnets for Waterbirds, and DistilBerts for SNLI.
>
> **Q5. *"...number of environments needs to be fixed ..."***
>
> **Response:** Yes, the number of environments (i.e. number of classifier heads) should be pre-defined beforehand. However, as discussed in Section 4.2.2 and Figure 5, EDNIL is not sensitive to the specified number of environments.
>
>
> **Q6. *"How is the environment decided at test time ..."***
>
> **Response:** As described in the Section 1 to 3, our ultimate goal is to learn an invariant model $M_\text{IL}$ ($X \rightarrow Y$) with robust performance at test time. The functionality of multi-head model $M_\text{EI}$ is to infer environments for the requirement of invariant learning algorithm. Therefore, $M_\text{EI}$ only serves during train time, and there is no need to determine the environment (or head) for prediction at test time.
>
> [1] Arjovsky, Martín et al. “Invariant Risk Minimization.” ArXiv abs/1907.02893 (2019).
>
> [2] Creager, Elliot et al. “Environment Inference for Invariant Learning.” ICML (2021).
>
> [3] Liu, Jiashuo et al. “Heterogeneous Risk Minimization.” ICML (2021).
>
> [4] Liu, Jiashuo et al. “Kernelized Heterogeneous Risk Minimization.” NeurIPS (2021).
>
> [5] Lin, Yong et al. “ZIN: When and How to Learn Invariance by Environment Inference?” ArXiv abs/2203.05818 (2022).

---

> > ### Comment · Reviewer_8Cwm · 2022-08-08
> > **Useful answers and much clearer explanation in updated paper version -> improved score and willing to improve more**
> >
> > Dear authors, thank you for your clarifications. I should say that my negative evaluation of your paper largely stemmed from my misunderstanding of your set-up. Your answers together with your updated version of the paper (which is now indeed a lot clearer and easier to follow) helped me to understand better. I have therefore improved my score and I am willing to improve even more upon clarification of some further concerns listed below.
> >
> > 1. $\mathcal{E}_{learn}$ definition is now missing in section 2.2.
> > 2. Lines 89-92: what do you mean by diversity? Please clarify it in the text.
> > 3. Line 96: the sentence does not seem to make quite sense in English. Something missing? Eg ... "respective ideas FOR identifying environments"?
> > 4. Please spell out the EIIL acronym at least once somewhere early in the text (will help the reader understand the EI and IL parts).
> > 5. I am still not clear about the role of the multiple heads in the EI model. Are these to the predictive functions for each environment? One head ~ one environment?
> > 6. What parts of the model get updated through optimizing $L_{EI}$? $f^{e}$ and/or $\Psi$? If yes, where do these enter $L_{IP}$ which is a variance of prediction accuracy of the invariant predictor $\Phi$ across all environments. Or is this loss used for updating something else?
> > 7. line 164: What do you mean by "to some extent"? Please elaborate.
> > 8. What is the meaning of "asterisk" and "dagger" in Table 3 and 4? ($KerHRM^*$ and EDNIL OOD result with $\dagger$.)
> > 10. Section 4.1 uses MLP for which network(s)? All or some/which ones?

---

> > > ### Author Response · Authors · 2022-08-09
> > > **Response to feedback**
> > >
> > > We thank Reviewer 8Cwm for the feedback about our responses and the revised paper. Below we discuss the questions the reviewer raised:
> > >
> > > **Q1. *"$\mathcal{E}_\text{learn}$ definition is now missing in section 2.2"***
> > >
> > > **Reponse:** Thank you for the reminder. We have added the definition of $\mathcal{E}_\text{learn}$ in Section 2.2 in the revised paper.
> > >
> > >
> > > **Q2. *"Lines 89-92: what do you mean by diversity? ..."***
> > >
> > > **Response:** As described in Line 91 (or Line 90 in the previous version), the diversity of environments represents the discrepancy of spurious correlations between environments. In practice, large discrepancy is favored for IRM.
> > >
> > > Take the case of two environments for example. When the spurious correlations in the two environments are 1 and -1 respectively, the diversity is desirable for giving a clear indication of the variant feature. On the other hand, when the spurious correlations are 0.5 and 0.5, the diversity is insufficient for identifying the variant feature.
> > >
> > > We have added the elaboration in line 90 to 96 in the paper.
> > >
> > > **Q3. *"Line 96: ..."***
> > >
> > > **Response:** Thank you for the reminder. We have rephrased the sentence in the revised paper.
> > >
> > > >*Here we provide a detailed introduction of the existing unsupervised methods identifying environments.*
> > >
> > >
> > > **Q4. *"Please spell out the EIIL acronym ..."***
> > >
> > > **Response:** Agree and done (Line 101). Thank you for the suggestion.
> > >
> > > **Q5. *"...the role of the multiple heads in the EI model..."***
> > >
> > > **Response:** Yes. Each head in the EI model represents one environment / distribution. In other words, each head is supposed to capture the relationships between variant features and the target in each environment.
> > >
> > > **Q6. *"What parts of the model get updated through optimizing $L_\text{EI}$? ... If yes, where do these enter $L_\text{IP}$ ..."***
> > >
> > > **Response:** As described in Line 175 (or Line 170 in the previous version), optimization of $L_\text{EI}$ is used to update $M_\text{EI}$, which consists of $f^e$ and $\Psi$.
> > >
> > > For $L_{IP}$, as we consider “soft” environment assignments when calculating expected loss of the invariant predictor, the gradient can be propagated back to $\Psi$ and $f^e$, who are involved in estimating the probabilities of environment assignment (Equation 5).
> > >
> > >
> > > **Q7. *"line 164: What do you mean by "to some extent"? ..."***
> > >
> > > **Response:** It means “after several training steps”. In each task / dataset, the actual training steps are determined by hyperparameters for the joint optimization.
> > >
> > > We have modified the sentence to improve the readability in the revised paper. Thank you for pointing this out.
> > >
> > > **Q8. *"...the meaning of "asterisk" and "dagger" in Table 3 and 4?..."***
> > >
> > > **Response:** For “asterisk”, as described in Line 214 (or Line 209 in the previous version), it is the average of top-5 scores selected from 10 runs for KerHRM. For “dagger”, as described in Line 207 (or Line 202 in the previous version), it is a sign indicating that the superiority is statistically significant (p-value less than 0.05) comparing with the 2nd-best result.
> > >
> > > **Q9. *"Section 4.1 uses MLP for which network(s)? ..."***
> > >
> > > **Response:** Both $M_\text{EI}$ (multi-head network) and $M_\text{IL}$ share the same architecture of the neural network, except for the output dimensions. That is, MLP is used for both $M_\text{EI}$ and $M_\text{IL}$ in Section 4.1.

---

### Meta-Review · Area_Chair_XLbd · 2022-08-25

**Recommendation:** Accept
**Confidence:** Certain

**Metareview:**

This work presents a novel environment-free invariant learning method that uses an auxiliary network to learn environment-specific features, from which environment inferences can be derived. The method is composed of two jointly learned models, that take care of the environment identification, the learning of the invariant representations, and the label predictions, produced by a multi-headed neural network. The proposed model is compared to different alternative models from the literature of the field, in different challenging benchmarks, and the results show that it closely achieves the best possible invariant learning performance.

After some initial discussions, all reviewers agreed that this work is ready for publication, as the work addresses an important problem, presents good empirical results, and will be of significant interest to the community.


**Award:**

No

---

### Decision · Program_Chairs · 2022-09-14

Accept